# Learning Robust Representations for Visual Reinforcement Learning via Task-Relevant Mask Sampling

**Vedant Dave**  *vedant.dave@unileoben.ac.at*
*Chair of Cyber-Physical-Systems*
*Technical University of Leoben, Austria*

**Ozan Özdenizci**  *oezdenizci@tugraz.at*
*Institute of Machine Learning and Neural Computation*
*Graz University of Technology, Austria*

**Elmar Rueckert**  *rueckert@unileoben.ac.at*
*Chair of Cyber-Physical-Systems*
*Technical University of Leoben, Austria*

**Reviewed on OpenReview:** *https://openreview.net/forum?id=2rxNDxHwtn*

## Abstract

Humans excel at isolating relevant information from noisy data to predict the behavior of dynamic systems, effectively disregarding non-informative, temporally-correlated noise. In contrast, existing visual reinforcement learning algorithms face challenges in generating noise-free predictions within high-dimensional, noise-saturated environments, especially when trained on world models featuring realistic background noise extracted from natural video streams. We propose Task Relevant Mask Sampling (TRMS), a novel approach for identifying task-specific and reward-relevant masks. TRMS utilizes existing segmentation models as a masking prior, which is subsequently followed by a mask selector that dynamically identifies subset of masks at each timestep, selecting those most probable to contribute to task-specific rewards. To mitigate the high computational cost associated with these masking priors, a lightweight student network is trained in parallel. This network learns to perform masking independently and replaces the Segment Anything Model (SAM)-based teacher network after a brief initial phase ($< 10 - 25\%$ of total training). TRMS enhances the generalization capabilities of Soft Actor-Critic agents under distractions, achieves better performance on the RL-Vigen benchmark, which includes challenging variants of the DeepMind Control Suite, Dexterous Manipulation and Quadruped Locomotion tasks.

## 1 Introduction

Visual Reinforcement Learning (RL) has garnered considerable success in mastering complex behaviors derived directly from high-dimensional, image-based observations (Mnih et al., 2015; Levine et al., 2016; Lee et al., 2020a; Laskin et al., 2020b). Conventionally, it is presumed that environmental observations, often obtained through hand-crafted features, contain only task-relevant information (Ha & Schmidhuber, 2018; Hafner et al., 2020; 2021; Hansen et al., 2024). This assumption enables RL algorithms to function in controlled settings with maximum efficiency, as it eliminates exogenous noise (irrelevant or uncontrollable external factors), such as weather fluctuations or random background movements, that could disrupt or impede the learning process. In the real world, the landscape is vastly different, rich in complex visual information, much of which is irrelevant to a specific task. The true challenge lies in accurately distinguishing task-relevant data while avoiding the unnecessary modeling of exogenous noise. Traditional RL approaches often fails to provide robust representations under noise, consequently failing to generalize. As a result,

they inadvertently incorporate irrelevant data into their representations, leading to the modeling of noise dynamics.

Recent approaches have sought to address this by selectively masking irrelevant parts of the input to focus learning on relevant information. Focus then Decide (FTD) by Chen et al. (2024) leverages the Segment Anything Model (SAM) (Kirillov et al., 2023) to select relevant segments via attention scores and attains high rewards. However, using SAM throughout the training process incurs high computational costs, making FTD impractical for complex tasks that require numerous masks. Contrary to this, methods like SGQN (Bertoin et al., 2022) use binarized attribution maps as masks to enforce consistency between the Q-values of masked and original images, highlighting relevant areas, but provides sub-optimal rewards and is extremely sensitivity to hyperparameter choices. InfoGating (Tomar et al., 2023) focuses only on offline RL experiments, using a multi-step inverse dynamics model and U-Nets (Ronneberger et al., 2015) to mask irrelevant features. Finally, SAM-G (Wang et al., 2023) employs the SAM model but depends on human intervention for mask selection. However, it remains unclear how combining multiple encoders yields keypoints for task-relevant masks.

To achieve robust performance and yield high rewards in noise-prone environmental settings, we propose TRMS, a novel algorithm that leverages existing masking techniques to learn task-relevant masks and to filter out irrelevant segments. This approach improves agent's generalization capabilities with respect to noise in multiple environments. The core idea of TRMS lies in employing pre-existing masking algorithms that extract meaningful substructures within an image by segmentation. Much like how the human brain processes visual information: decomposing scene into objects and selectively attending to task-relevant elements (Kaiser et al., 2016; Seidl et al., 2012a; Peters & Kriegeskorte, 2021), our method focuses on high-level abstractions, bypassing pixel-level relevance assessments. By segmenting the image into semantically meaningful subregions, we substantially reduce the complexity of the selector's task, requiring it to identify relevance from a more refined subset of the scene. This approach sharply contrasts with methods that attempt to infer such abstractions from raw pixel data (Bertoin et al., 2022; Hansen et al., 2021; Grooten et al., 2024), which inevitably suffer from less efficient learning (as shown in the evaluations through empirical results). Instead of evaluating every pixel individually, our method simplifies the process to determining whether a mask (representing a smaller subset of pixels) is correct or not. A selector network, utilizing a Convolutional Neural Network (CNN), provides a binary output for each mask, classifying it as relevant or irrelevant.

To improve the computational efficiency, we include the student network in our training procedure. The pre-trained segmentation model remains frozen throughout training and is used solely to generate output masks during inference. The approach is executed in two key phases: (i) Student Network is used to learn the mask generation over a subset of the batch to mitigate the overhead of processing the entire batch. (ii) After $T_{\text{train}}$ steps, the student network, a lightweight CNN architecture, replaces the teacher network, enabling faster computations. An empirical evaluation of wall time is shown in Appendix D. Since the student network is co-optimized with the encoder, no auxiliary loss function is required beyond the initial masking loss during the first phase. The encoder's loss alone is sufficient to guide the optimization process.

To evaluate the performance of TRMS, we conducted experiments across modified version of **eleven** environments from three benchmarks in RL-ViGen (Yuan et al., 2023): the Deepmind Control Generalization Benchmark (Hansen et al., 2021), Quadruped Locomotion (Hansen et al., 2021) and Dexterous Manipulation (Rajeswaran et al., 2018). Moreover, TRMS outperforms **eight** well-established methods in various tasks in vision-based reinforcement learning (Hansen et al., 2021; Yuan et al., 2022b; Huang et al., 2022; Bertoin et al., 2022; Yarats et al., 2021; 2022; Laskin et al., 2020a; Wang et al., 2022), achieving better performance across multiple environments.

Our main contributions are:

- We propose TRMS, a novel algorithm with an actor-critic architecture that enhances task-relevant masking by leveraging pre-trained segmentation for providing semantically meaningful subregions, with a selector that identifies task-relevant information from these subregions, improving both generalization and robustness in visually complex environments.

- We further optimize TRMS through a two-phase training process, where a lightweight student network incrementally replaces a teacher network, enabling faster mask learning and improving computational efficiency as compared to solely relying on heavy segmentation model (SAM).

- We validate TRMS through comprehensive experiments across eleven modified environments from the RL-ViGen benchmarks, where it surpasses relevant existing methods in most of the environments in vision-based reinforcement learning tasks.

## 2 Related Work

**Generalization in Visual RL.** RL agents struggle with severe generalization limitations, where performance degrades sharply in unfamiliar environments due to overfitting and insufficient adaptability to unseen variations (Kirk et al., 2023; Jiang et al., 2023; Raileanu et al., 2021; Zhang et al., 2018; Cobbe et al., 2019). Numerous methods have been developed to improve generalization in reinforcement learning, including domain adaptation (Xing et al., 2021b; Li et al., 2022; Sun et al., 2022), domain randomization (Mehta et al., 2020; Lee et al., 2020b; Tobin et al., 2017), and curriculum learning (Narvekar et al., 2020; Gupta et al., 2022). Contrastive learning (Laskin et al., 2020a; Agarwal et al., 2021; Liu et al., 2023a; Yang et al., 2022), bisimulation metrics (Ferns et al., 2011; Zhang et al., 2021; Liu et al., 2023b; Sun et al., 2024; Zang et al., 2022), data augmentations (Hansen & Wang, 2021; Laskin et al., 2020b; Raileanu et al., 2021; Yarats et al., 2021; 2022; Mumuni & Mumuni, 2022), keypoints (Wang et al., 2021; 2023), and information-theoretic approaches (Tomar et al., 2023; Fan & Li, 2022; Dave & Rueckert, 2024; You et al., 2022; Wang et al., 2024) improve state representations, whereas imitation learning builds policies robust to perturbations (Fan et al., 2021; Xing et al., 2021a; Wang & Hager, 2024).

Numerous works have proposed different ideas to mitigate the impact of task-irrelevant distractors in reinforcement learning environments (Yarats et al., 2022; Hansen et al., 2021; Huang et al., 2022; Laskin et al., 2020a; Yang et al., 2023; Yuan et al., 2022a; Wang et al., 2024). SODA (Hansen & Wang, 2021) incorporates a BYOL-like (Grill et al., 2020) architecture and augments data by linearly combining supplementary images with observations. TLDA (Yuan et al., 2022a) takes a different approach, recommending the exclusion of task-critical pixels from augmentation, determined through the use of Lipschitz constants. PIEG (Yuan et al., 2022b) employs a pre-trained ResNet (He et al., 2016) as its backbone for generalization under distractors. SRM (Huang et al., 2022) learns representations in frequency-domain and learns to discard certain frequency in the observation to address domain shifts. CG2A (Liu et al., 2023c) identifies potential conflicts among gradients generated by different augmentations and investigates how to better integrate these augmentations.

**Masking Distractors in RL.** Several approaches have been proposed to enhance the generalization of RL agents by selectively masking parts of the input. SGQN (Bertoin et al., 2022) proposes a saliency-guided method, where binarized attribution maps serve as input masks. It regularizes the value function by enforcing consistency between the Q-values of the masked and original state images, improving learning focus on relevant areas. MLR (Yu et al., 2022) introduces a self-supervised auxiliary objective that performs random masking and reconstructs masked information in the latent space, encouraging dynamic-relevant state representations. InfoGating (Tomar et al., 2023) utilizes a multi-step inverse dynamics model as its primary objective and employs U-Nets to mask irrelevant information, with a focus on offline RL experiments. MaDi (Grooten et al., 2024) closely aligns to our approach in utilising a small CNN for masking (similar to our student network) and using reward-driven supervision to suppress irrelevant pixels, but learns masks end-to-end without leveraging structured segmentation priors. SAM-G (Wang et al., 2023) employs a frozen Segment Anything Model (SAM) (Kirillov et al., 2023) model to generate observation masks. However, it fundamentally differs in its reliance on foundation-model prompts, necessitating human intervention for mask selection. Moreover, the mechanism by which multiple encoders are combined to yield task-relevant masks remains unclear. FTD (Chen et al., 2024) also uses the SAM to select relevant segments via attention scores and regularizes the RL method with inverse dynamics and reward loss. However, the use of SAM throughout the training process results in high computational costs, making it impractical for complex tasks that require numerous masks. These methods rely on unstructured mask learning, costly vision models, or lack spatial selectivity. In contrast to these methods, TRMS combines segmentation priors from SAM with a dynamic selector and lightweight student network.

**Relation to Cognitive Perception.** Selecting visual information from cluttered real-world scenes requires aligning visual input with the observer's attentional set—an internal representation of objects relevant to current behavioral goals—and as these goals shift, a new attentional set must be instantiated, necessitating the suppression of the previous set to prevent distractions from irrelevant objects (Nasr et al., 2008; Seidl et al., 2012b). Segmentation allows for the tracking and prediction of objects, enabling cognitive functions like memory and action planning independent of sensory input (Scholl, 2001; Brady et al., 2011). Processes such as perceptual grouping, proto-objects, and object files underpin how humans segment and recognize relevant objects in complex scenes (Roelfsema & Ooyen, 2005; Gao et al., 2016). Inspired by human-like segmentation and attention mechanisms, we introduce a segment sampling strategy that leverages masking priors to learn robust and task-relevant representations.

## 3 Preliminaries

### 3.1 Visual Reinforcement Learning

Reinforcement Learning (RL) aims to obtain optimal policies for sequential decision-making problems through iterative interactions with the environment (Sutton, 2018). In contexts where agents receive high-dimensional sensory inputs, such as visual observations, these problems are aptly modeled as Partially Observable Markov Decision Processes (POMDPs). A POMDP is formally defined by the tuple $\mathcal{M} = \langle \mathcal{S}, \mathcal{A}, \mathcal{O}, P, \Omega, R, \gamma \rangle$, where $\mathcal{S}$ is the set of latent states, $\mathcal{A}$ is the action space, $\mathcal{O}$ is the observation space, $P : \mathcal{S} \times \mathcal{A} \to \mathcal{P}(\mathcal{S})$ is a state transition function, $\Omega : \mathcal{S} \to \mathcal{P}(\mathcal{O})$ is the observation function, $R : \mathcal{S} \times \mathcal{A} \to \mathbb{R}$ is the reward function, and $\gamma \in [0, 1)$ is the discount factor, which attenuates future rewards. To mitigate the challenges associated with partial observability (Kaelbling et al., 1998), we redefine the agent's state $s_t$ as a sequence of $k$ consecutive observations, namely $s_t = (o_t, o_{t-1}, \ldots, o_{t-k+1})$, where each $o_i \in \mathcal{O}$. Although this window does not capture the full action-observation history, it often serves as a practical surrogate in visual domains, enriching the agent's perceptual input and partially recovering temporal dependencies lost in single-frame observations. The agent's objective is to obtain a policy $\pi_{\phi_a}$, parameterized by $\phi_a$, that maximizes the expected cumulative discounted return: $\mathbb{E}_{\kappa \sim \pi_{\phi_a}} [\sum_{t=0}^{\infty} \gamma^t R(o_t, a_t)]$, where the trajectory $\kappa$ is induced by the underlying dynamics of the POMDP.

### 3.2 Soft Actor-Critic (SAC)

Our methodology builds upon the Soft Actor-Critic (SAC) algorithm (Haarnoja et al., 2018), a model-free, off-policy RL approach that integrates entropy maximization into the policy optimization framework to enhance exploration and stabilize learning. SAC employs a critic network $Q_{\theta_q}$ to approximate the soft state-action value function, seeking to estimate the optimal action-value function $Q^*(s, a)$ in the context of stochastic policies. The actor is instantiated as a stochastic policy $\pi_{\phi_a}$ that aims to maximize both the expected return and the entropy of the policy, thereby encouraging exploration of the action space. The shared encoder maps the high-dimensional observation space $\mathcal{O}$ into a lower-dimensional representations. To ensure the stability of the learning process, the critic and shared encoder have target networks start with the same parameters $\theta_{tgt} = \theta_q$. These target networks are updated via an exponential moving average (EMA), $\theta_{tgt} \leftarrow (1 - \epsilon)\theta_{tgt} + \epsilon\theta_q$, where $\epsilon \in (0, 1)$. The EMA update serves to temper abrupt fluctuations in parameter values, thereby contributing to the stability of the training process.

### 3.3 Generalization in Visual RL

Our work focuses on the challenge of generalization in visual reinforcement learning, where the agent is trained on the environment without distractors (including augmentations) and then evaluated on previously unseen environment with distractors. The goal is to obtain consistent behaviour under domain (environment distribution) shift. Formally, we consider a family of POMDPs, denoted by $\mathcal{M} = \{\mathcal{M}_1, \mathcal{M}_2, ..., \mathcal{M}_k\}$. Each POMDP $\mathcal{M}_i \sim \mathcal{M}$ shares the same underlying dynamics and reward structure but differs in its observation space $\mathcal{O}_i$, typically due to variations in visual appearances. Our objective is to learn a policy $\pi$ that maximizes the expected cumulative return across POMDPs sampled from $\mathcal{M}$ in a zero-shot manner i.e.

without additional training or fine-tuning on the test environments. The goal is to find a policy $\pi$ that maximizes the expected discounted return: $\eta_{\mathcal{M}}(\pi) = \mathbb{E}_{(o_t, a_t) \sim (\mathcal{M}, \pi)} \left[ \sum_{t=0}^{T-1} \gamma^t R(o_t, a_t) \right]$. We denote the training environment as $\mathcal{M}_{\text{train}}$ and the set of test environments as $\mathcal{M}_{\text{test}}$. The generalization performance of the policy can be quantified by the generalization gap (Kirk et al., 2023; Wang et al., 2024), defined as $\mathcal{L}_{\text{gen}} = \eta_{\mathcal{M}_{\text{test}}}(\pi) - \eta_{\mathcal{M}_{\text{train}}}(\pi)$.

## 4 Method

We propose Task-Relevant Mask Sampling (TRMS), an algorithm designed to enhance generalization and robustness in noise-prone environments by masking out irrelevant segments. In this section, we provide a detailed overview of TRMS's architectural design, training procedure and its inclusion within the reinforcement learning framework.

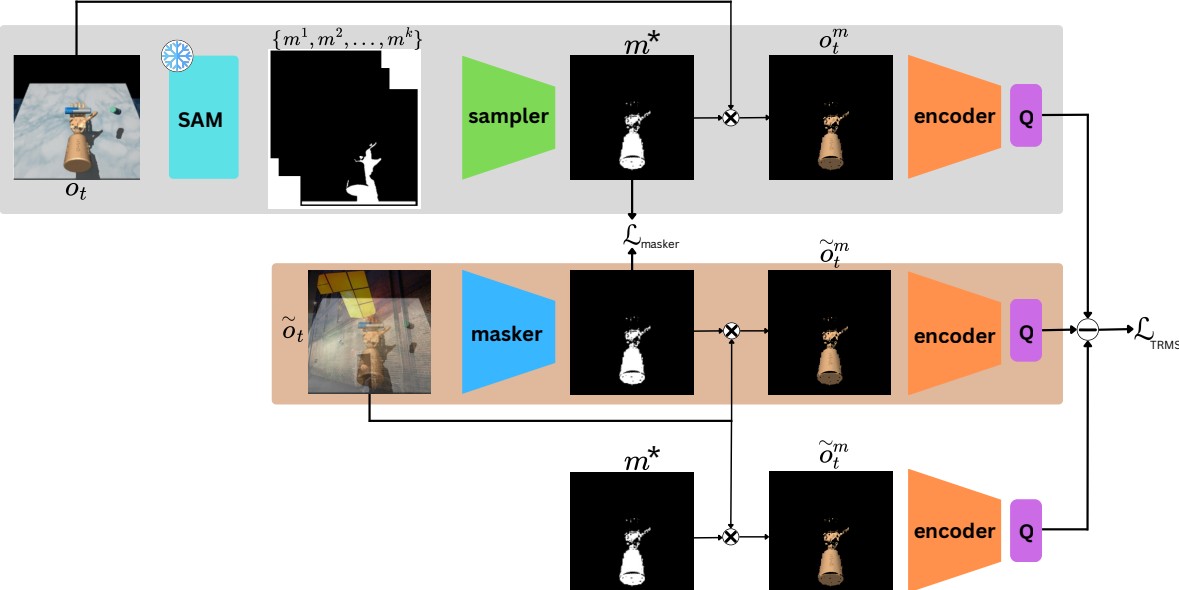

Figure 1: Depiction of the Task-Relevant Mask Sampling (TRMS) architecture, detailing its components and sequential training phases. Initially, $k$-masks are generated by a frozen segmentation model, followed by sampling and a logical OR operation to produce a single mask $m^*$. This mask is then applied to the non-augmented image via a Hadamard product $\otimes$, producing the masked image $o_t^m$, which progresses through the encoder and $Q$-network to yield the corresponding $q$-value. Simultaneously, $m^*$ is utilized to mask the augmented image $\tilde{o}_t^m$ through the Hadamard operation. During the initial teaching phase, $T_{\text{Teach}}$, only the row in ▨ is activated. Beyond this phase, the student masking network replaces the segmentation and sampler, and is trained utilizing the ▨ row.

### 4.1 Masking Prior

TRMS explicitly leverages existing pre-trained segmentation models as a backbone to extract masks from non-augmented images. In this case, we employ the Fast Segment Anything Model (FastSAM) (Zhao et al., 2023)[1], known for its compact design that reduces memory usage and provides fast inference. As shown in Figure 1, this network remains frozen throughout the training phase, operating solely as an inference model.

Let $o_t$ and $o_{t+1}$ denote the observation at time step $t$ and $t+1$ respectively. Following the approach in existing methods (Hansen & Wang, 2021), we apply an augmentation $\psi$ by overlaying a random image onto these observations, resulting in augmented pairs $\{\tilde{o}_t, \tilde{o}_{t+1}\} = \{\psi(o_t), \psi(o_{t+1})\}$. Since these augmented images

---

[1]As FastSAM is a variant of SAM, here we use FastSAM and SAM interchangably.

hinder accurate mask extraction from SAM, we rely on non-augmented images for precise and consistent masks (as shown in ▨ in Fig 1). Provided an non-augmented image $o$, we utilize a frozen mask extractor (SAM), denoted by $M$, to obtain $k-$masks, $\{m^1, m^2, ..., m^k\}$.

## 4.2 Mask Sampling Network

These $k$ masks are fed into a CNN-based mask selection module, denoted as $G_\beta(m)$, which generates the probability of selection for each mask $w(m^i)$. The goal of the mask selector is to identify task-relevant masks, specifically those with selection probabilities exceeding the threshold $1/k$ and to disregard all others. Formally, this selection is represented as follows:

$$p(m^i) = \begin{cases} 1, & \text{if } w(m^i) \geq 1/k, \\ 0, & \text{otherwise.} \end{cases} \tag{1}$$

The objective of the mask selector is to encourage the probabilities associated with task-relevant masks to exceed $1/k$. Subsequently, we compute the Hadamard product across all selected masks, resulting in a final composite mask. This mask is then applied to both the original and augmented images, isolating the task-relevant regions of the observations,

$$m^* = \bigvee_{i=1}^{k} \left[ p(m^i) \cdot m^i \right], \tag{2}$$

where $\bigvee$ denotes the Logical OR operation. Following this, the Hadamard product is calculated with the original and augmented observations, respectively, as follows,

$$o_t^m = m^* \otimes o_t, \tag{3}$$

$$\tilde{o}_t^m = m^* \otimes \tilde{o}_t, \tag{4}$$

where $\otimes$ represents the Hadamard product. $o_t^m$ and $\tilde{o}_t^m$ are the images obtained by applying the same sampled masks on non-augmented and augmented observations. This process is applied to the observations at both time steps, $t$ and $t+1$, ensuring that the task-relevant features are preserved across temporal frames for both the original and augmented observations.

Multiple masks may be relevant to the task; thus, the selector must be capable of selecting several masks simultaneously. Empirical observations suggest that using only a Softmax activation function at the output layer often results in higher selection probabilities for only a few masks (typically one or two). However, complex scenes frequently require the use of multiple masks. To address this limitation, we employ the Gumbel-Softmax distribution (Gumbel, 1954; Jang et al., 2017) at the output layer, which helps mitigate this selection bias. By adjusting the temperature parameter $\tau$, we can encourage a more uniform probability distribution, making it easier for the selector by requiring only a slight increase in probability for the relevant masks to be chosen. The Gumbel-Softmax is defined as

$$y_i = \frac{\exp\left((\log(x_i) + g_i)/\tau\right)}{\sum_{j=1}^{k} \exp\left((\log(x_j) + g_j)/\tau\right)}, \tag{5}$$

where $g_i$ are i.i.d. samples from a Gumbel distribution, typically computed as $g_i = -\ln(-\ln(u_i))$ with $u_i \sim \text{Uniform}(0, 1)$. This formulation enables more robust selection in complex scenes with multiple relevant areas by reducing the chances of a single mask's probability dominating the distribution. More details are provided in the Appendix A. Since we perform a thresholding operation in Eq. (1), we apply Straight-Through Estimators (STE) to address the backpropagation challenges associated with discrete operations (Bengio et al., 2013).

## 4.3 Q-relevant Mask Sampling

To effectively select task-relevant masks, we assume that the task is already well-performed in the original, non-augmented environment. If the algorithm cannot solve the task without augmentation, then solving it

would be infeasible. Therefore, we consider the $Q$-values obtained from the non-augmented environment as expert $Q$-values. For each pair of masked augmented and non-augmented observations, denoted as $o^m$ and $\tilde{o}^m$, we obtain their corresponding state representations $s$ and $\tilde{s}$. The target q function can be defined as $q_{\text{tgt}} = r(s_t, a_t) + \gamma \max_{a'_t} Q_{\text{tgt}}(s_{t+1}, a'_t)$. The $Q$-loss function for the non-augmented images is then given by

$$\mathcal{L}_Q(\theta_q, \beta) = \mathbb{E}_{(s_t, a_t, r_t, s_{t+1}) \sim \mathcal{B}} \left[ \frac{1}{2} \left( q_{\text{tgt}} - Q_{\theta_q}(s_t, a_t) \right)^2 \right], \tag{6}$$

where $\mathcal{B}$ denotes the replay buffer. In a similar way, the q-target for the augmented images can be written as $q_{\text{tgt}} = r(\tilde{s}_t, a_t) + \gamma \max_{a'_t} Q_{\text{tgt}}(\tilde{s}_{t+1}, a'_t)$ and the corresponding $Q$-loss is

$$\mathcal{L}_{\tilde{Q}}(\theta_q, \beta) = \mathbb{E}_{(s_t, a_t, r_t, s_{t+1}) \sim \mathcal{B}} \left[ \frac{1}{2} \left( \tilde{q}_{\text{tgt}} - Q_{\theta_q}(\tilde{s}_t, a_t) \right)^2 \right]. \tag{7}$$

The Task-Relevant Mask Selection (TRMS) loss is defined as the average of these two losses:

$$\mathcal{L}_{\text{TRMS}}(\theta_q, \beta) = \frac{1}{2} \left( \mathcal{L}_Q(\theta_q, \beta) + \mathcal{L}_{\tilde{Q}}(\theta_q, \beta) \right). \tag{8}$$

This TRMS loss simultaneously optimizes the critic's accuracy over both augmented and non-augmented observations, thereby promoting robustness to input augmentations when identical masks are applied. If task-irrelevant segments are selected, the loss will increase. By discarding visually irrelevant segments, the mask selector reduces representational variance from distractor pixels and encourages the policy to focus on task-relevant features. Consequently, this allows networks to consistently observe the same pixel value across different augmentations, thus maintaining a coherent state representation regardless of the augmentation applied. The Total Loss, a combination of TRMS and masking loss ($\mathcal{L}_{\text{masker}}$ in Eq. (10)), of the overall architecture can be defined as

$$\mathcal{L}_{\text{Total}} = \mathcal{L}_{\text{TRMS}} + \mathcal{L}_{\text{masker}}. \tag{9}$$

### 4.4 Student Masking Networks

Although it is technically feasible to run the algorithm for 500K steps using the FastSAM models and a mask sampler, this approach is highly computationally intensive due to the considerable time required by the masking models. These models, while accurate, are not optimized for speed and may impose significant delays. Our empirical results shows that the model utilizing only SAM can take upto 3 days on the *Walker Walk* task (Appendix D). Moreover, FTD (Chen et al., 2024), which also relies on SAM, can require up to 3-6 days to process a single seed, depending on both the task complexity and the number of masks to be generated. To mitigate this computational burden, we introduce a Student Masking Network, a CNN-based network that effectively mimics the behavior of the Prior Masking model but only for a defined initial period, denoted as $T_{\text{teach}}$. During this period, the student network learns directly from the teacher model, replicating its outputs.

The training objective for this student network capitalizes on the binary nature of the mask outputs. We employ a Binary Cross-Entropy (BCE) loss to measure the discrepancy between the teacher network's output mask, $m^*$ (as defined in Eq. (2)), and the mask generated by the student network, $M_\alpha^*(o_t)$, which is parameterized by $\alpha$. Formally, this is expressed as

$$\mathcal{L}_{\text{masker}} = \text{BCE}(m^*, M_\alpha^*(\tilde{o}_t)). \tag{10}$$

This BCE loss is then exclusively backpropagated through the student network, enabling it to gradually learn the teacher's masking strategy during the initial training period $T_{\text{teach}}$. After completing these $T_{\text{teach}}$ steps, the teacher network is omitted, and only the student network is utilised to generate task-relevant masks, maintaining operational efficiency while significantly reducing computation time (Appendix D). An additional strategy to further optimize training time is implemented after $T_{\text{teach}}$, we increase the batch size. This adjustment expedites the learning process by enabling the student network to process more data per

---

**Algorithm 1** Training Algorithm for SAC with TRMS

---

**Require:** $E_{step}, \psi, k, \lambda_q, \lambda_\pi, \lambda_\alpha$      ▷ Variables Initialization
**Require:** $\phi_a, \theta_q, M, M_\alpha^*, G_\beta$      ▷ Networks Initialization
1:  $D \leftarrow \emptyset$      ▷ Initialize replay buffer
2:  **for** each initial collection step **do**
3:     $a_t \sim \pi_{random}(\cdot|o_t)$      ▷ Sample random action
4:     $o_{t+1}, r_{t+1} \sim E_{step}(a_t)$      ▷ Apply action
5:     $D \leftarrow D \cup (o_{t+1}, a_t, r_{t+1})$      ▷ Append to buffer
6:  **end for**
7:  **for** every training step **do**
8:     $\{(o_t, a_t, r_t, o_{t+1})\}_{t=k}^{L+k} \sim D$      ▷ Sample minibatch
9:     $a_t \sim \pi_{\phi_a}(a_t|o_t)$      ▷ Sample action
10:     $o_{t+1}, r_t \sim E_{step}(a_t)$
11:     $D \leftarrow D \cup (o_t, a_t, r_t, o_{t+1})$
12:     $\tilde{o}_t, \tilde{o}_{t+1} \leftarrow \psi(o_t), \psi(o_{t+1})$      ▷ Augmentation
13:     **for** each gradient step **do**
14:         **if** step $\leq T_{teach}$ **then**
15:             $\tilde{o}_t^m, \tilde{o}_{t+1}^m \leftarrow \tilde{o}_t \otimes G_\beta(M(o_t)), \tilde{o}_{t+1} \otimes G_\beta(M(o_{t+1}))$      ▷ SAM
16:         **else**
17:             $\tilde{o}_t^m, \tilde{o}_{t+1}^m \leftarrow \tilde{o}_t \otimes M_\alpha^*(\tilde{o}_t), \tilde{o}_{t+1} \otimes M_\alpha^*(\tilde{o}_{t+1})$      ▷ CNN Masking
18:         **end if**
19:         Update Encoder and Mask Sampler (Eq. (8))
20:         $\theta_q \leftarrow \theta_q - \lambda_q \nabla \mathcal{L}_{TRMS}(o_t, o_{t+1}, \tilde{o}_t^m, \tilde{o}_{t+1}^m; \theta_q, \beta)$
21:         $\phi_a \leftarrow \phi_a - \lambda_\pi \nabla \mathcal{L}_\pi(\phi_a)$      ▷ Update policy
22:         $\alpha \leftarrow \alpha - \lambda_\alpha \nabla \mathcal{L}_{masker}(o_t, \tilde{o}_t; \alpha)$      ▷ Update Masker
23:     **end for**
24: **end for**

---

training iteration. Architectural details are provided in Appendix C. The details about the training of the entire architecture in Fig. 1 is provided in Algorithm 1[2]. We augmented the SAC algorithm with TRMS components, shown in blue.

## 5 Experiments

In this section, we present our experimental evaluations conducted on generalization benchmarks from RL-ViGen (Yuan et al., 2023). These benchmarks were selected as they encompass a wide range of environments: **(1)** DeepMind Control Generalization Benchmark (Hansen et al., 2021) for evaluating continuous control agents across tasks with complex dynamics and diverse rewards, essential for testing generalization capabilities; **(2)** Quadruped Locomotion (Hansen et al., 2021), which includes a Unitree quadruped robot, challenging agents with intricate balance and control requirements in dynamic environments; **(3)** Dexterous Manipulation (Rajeswaran et al., 2018), featuring multi-object interactions and sparse rewards that require precise control strategies to handle sophisticated manipulation tasks. We provide a comprehensive description of our experimental configurations and compare the performance of TRMS against relevant existing approaches. This analysis demonstrates the effectiveness of TRMS in enhancing generalization across diverse tasks in vision-based reinforcement learning. An evaluation of wall time comparing TRMS with the Only-SAM method is presented in Appendix D. The analysis demonstrates that incorporating the student network substantially enhances computational efficiency compared to relying solely on a heavily parameterised segmentation model.

---

[2]Temperature update in SAC, double critics and target network updates are omitted for clarity.

**Baselines.** Our method is compared against several prominent Visual RL algorithms that are specifically designed for generalization. DrQ (Yarats et al., 2021) improves SAC (Haarnoja et al., 2018) by augmenting visual inputs while updating the TD loss. DrQ-v2 (Yarats et al., 2022), a DDPG (Lillicrap et al., 2016) and DrQ-based model-free algorithm. CURL (Laskin et al., 2020a) enhances visual representations by using a contrastive learning approach similar to SimCLR (Chen et al., 2020) i.e. aligning augmented views of the same observation. SVEA (Hansen et al., 2021) stabilizes learning by using un-augmented images for the target Q-value, while applying augmentation to reduce Q-value variance. SRM (Huang et al., 2022) learns representations in frequency-domain and learns to discard certain frequency in the observation to address domain shifts. PIE-G (Yuan et al., 2022b) incorporates ResNet (He et al., 2016) pre-trained models to enhance generalization, while SGQN (Bertoin et al., 2022) uses saliency maps to focus on key pixels crucial for decision-making. VRL3 (Wang et al., 2022) is SOTA algorithm for Adroit tasks, utilizing human demonstrations.

**Augmentation.** All of our baselines, except for SAC, leverage specific data augmentation techniques during training. TRMS uses an augmentation strategy inspired by SVEA (Hansen et al., 2021), which has proven effective for handling distracting video backgrounds through overlay augmentation. In this approach, we overlay a randomly selected image $n$ from the Places dataset (Zhou et al., 2018) onto our observation frame as $\tilde{o}_t = \delta \cdot o_t + (1 - \delta) \cdot n$, where $\delta$ is a weighting factor that controls the degree of image overlay. In all the experiments, we set $\delta = 0.5$.

**Zero-shot Evaluation.** To assess generalization, we perform zero-shot evaluations of the trained agents on a range of unseen environments with different distraction intensities. Specifically, we evaluate performance on the video-easy and video-hard configurations across all environments. Each seed undergoes evaluation over 100 episodes, corresponding to the designated noise levels.

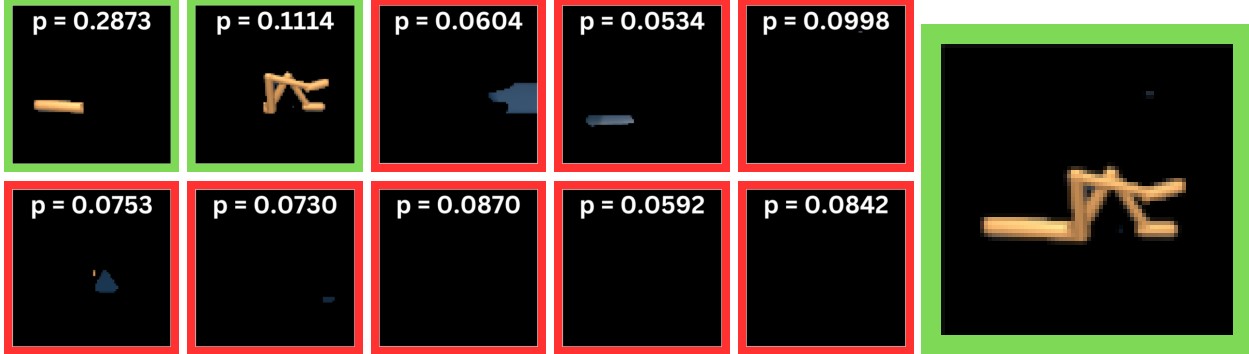

Figure 2: The figure on the left shows the probability of each mask out of the 10 masks provided. As there are 10 masks, only the masks above the probability of 0.1 (1/10) are selected, resulting into the image on the right.

## 5.1 Deepmind Control Suite

We evaluate our algorithm on the DMC-GB (Hansen & Wang, 2021) benchmark, spanning six tasks: Walker Walk, Walker Stand, Ball in Cup Catch, Finger Spin, Cartpole Swingup, and Cheetah Run. RL-ViGen integrates DMC-GB with two difficulty levels: video-easy (10 background videos) and video-hard (100 background videos without surface), where natural video backgrounds are used to rigorously test generalization under varying visual distractions.

**Generalization Performance.** We evaluate generalization by running five seeds per task and calculating the mean and standard deviation of returns. As shown in Table 1, TRMS surpasses all the baselines in 10 out of 12 environments. Notably, TRMS demonstrates an advantage in the video-hard setting, the most challenging environment due to its complex video perturbations, where it outperforms all the selected

Table 1: Performance on DMC Benchmark Environment in Video-Hard (VH) and Video-Easy (VE) settings. S-up: Swingup.

| Task (VE) | SAC | DrQ | DrQ-v2 | CURL | SVEA | SRM | PIEG | SGQN | TRMS | Δ |
|---|---|---|---|---|---|---|---|---|---|---|
| Cartpole S-up | 398±60 | 485±105 | 267±41 | 404±67 | **782±27** | 724±75 | 482±51 | 717±35 | **787±35** | +5 (0.63%) |
| Walker Walk | 245±165 | 682±89 | 175±117 | 556±133 | 819±81 | 854±42 | **871±22** | 860±53 | 863±74 | -8 (0.91%) |
| Walker Stand | 389±131 | 873±83 | 560±48 | 852±75 | 961±8 | **966±42** | 957±12 | 955±9 | **967±5** | +1 (0.10%) |
| Ball in Cup | 192±157 | 318±157 | 871±106 | 316±119 | 871±106 | 924±35 | 910±37 | 761±171 | **938±10** | +14 (1.15%) |
| Finger Spin | 206±169 | 533±119 | 456±15 | 502±19 | 808±33 | 853±76 | 837±107 | 609±61 | **868±24** | +31 (3.70%) |
| Cheetah Run | 87±21 | 102±30 | 64±22 | 104±24 | 249±20 | 257±21 | **287±20** | 269±33 | 207±83 | -80 (27.87%) |
| **Average** | 253 | 499 | 457 | 456 | 757 | 763 | 724 | 697 | **772** | +9 (1.18%) |
| Task (VH) | SAC | DrQ | DrQ-v2 | CURL | SVEA | SRM | PIEG | SGQN | TRMS | Δ |
| Cartpole S-up | 158±17 | 138±9 | 130±3 | 114±15 | 393±45 | 475±75 | 323±24 | 488±18 | **514±102** | +26 (5.33%) |
| Walker Walk | 122±47 | 104±22 | 34±11 | 58±18 | 377±93 | 535±35 | 641±63 | 655±45 | **747±63** | +92 (14.05%) |
| Walker Stand | 231±57 | 289±49 | 151±13 | 45±5 | 834±46 | 863±57 | 852±56 | 851±24 | **906±20** | +43 (4.98%) |
| Ball in Cup | 101±37 | 100±40 | 97±27 | 115±33 | 403±174 | 566±135 | 773±74 | 782±57 | **837±20** | +55 (7.05%) |
| Finger Spin | 13±10 | 91±13 | 21±4 | 27±21 | 335±58 | 419±32 | 762±59 | 554±8 | **791±54** | +29 (5.19%) |
| Cheetah Run | 10±5 | 32±13 | 23±5 | 21±7 | 105±37 | 115±24 | 154±17 | 144±34 | **189±86** | +35 (22.72%) |
| **Average** | 106 | 126 | 76 | 63 | 408 | 496 | 584 | 579 | **664** | +80 (13.70%) |

relevant baselines. In the video-easy setting, TRMS exhibits an improvement of 1.18% over the second-best performing method, SRM, and 5% over the average of the next four best methods (SVEA, SRM, PIEG and SGQN). Interestingly, in the video-hard settings, TRMS not only outperforms all baselines across every environment but also achieves an impressive average improvement of 13.70% over the second-best method, SGQN, as shown by the Δ in Table 1. Collectively, these results underscore TRMS's robust generalization capabilities, particularly under high-noise and complex video conditions, establishing it as a reliable solution across diverse test environments.

**Mask Sampling Probabilities Visualization.** The mask selection probability from the masker is illustrated in Figure 2. Given that there are 10 masks, the selector's objective is to increase the probability of task-relevant segments above the threshold of 1/10 (i.e., 0.1) while reducing the probability of irrelevant segments below this threshold. this case, only the first two segments are selected (i.e., only their masks, without the overlayed image). These selected segments are then subjected to the Logical OR operation and the Hadamard product, as depicted in Eq. (2) and Eq. (3) respectively. The image on the right of the figure represents the resulting output, clearly demonstrating the selective focus on relevant regions.

**Representations under Distractors.** To demonstrate that TRMS' capability of learning domain-invariant representations, we employ t-SNE (van der Maaten & Hinton, 2008) to visualize the features

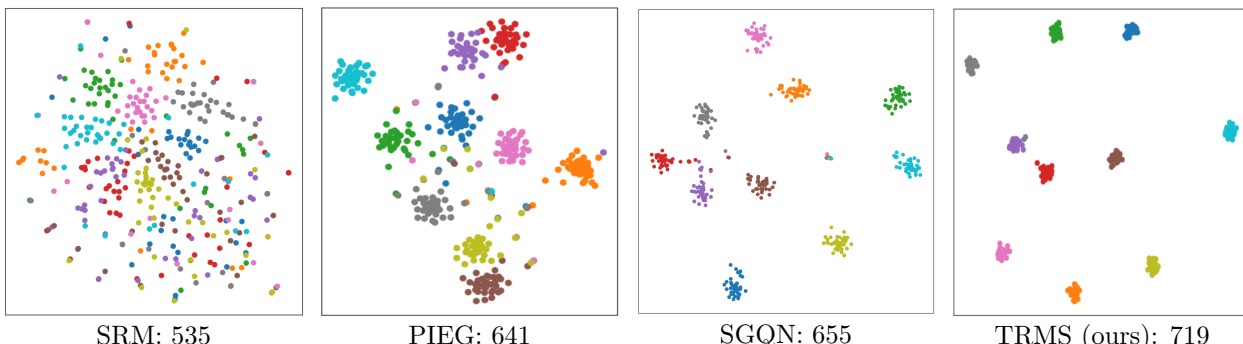

| SRM: 535 | PIEG: 641 | SGQN: 655 | TRMS (ours): 719 |

Figure 3: t-SNE visualization of clustering results for TRMS and three baselines. TRMS demonstrates a more distinct and well-separated clustering pattern, with each cluster representing identical agent poses with distinct backgrounds.

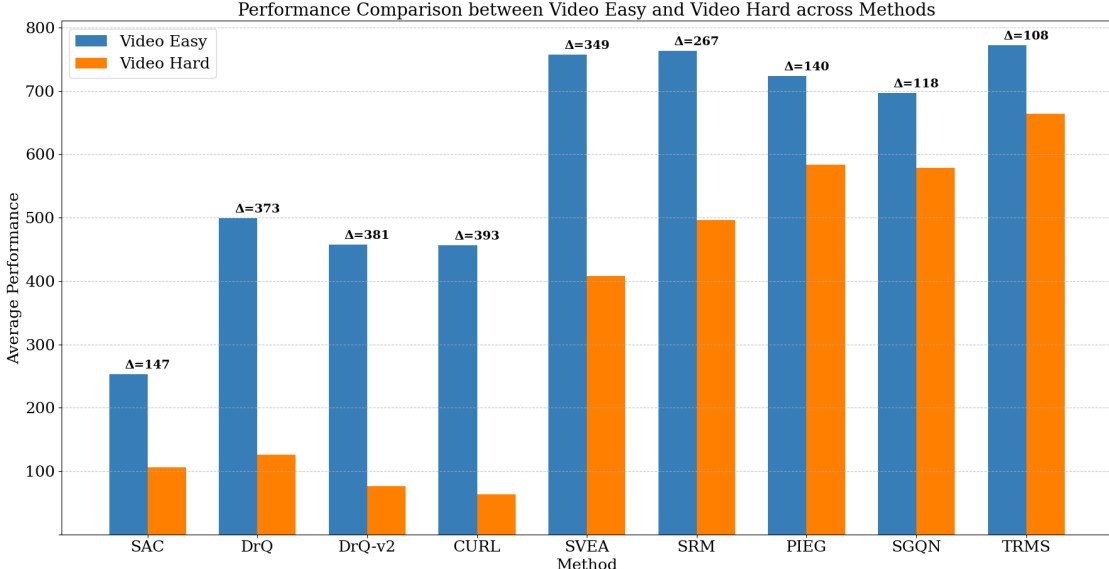

Figure 4: For each method, paired bars display the average performance under Video Easy and Video Hard conditions, and the annotated $\Delta$ indicates the absolute difference between the two settings. Standard deviations are omitted, as the underlying tasks differ significantly, making such variance comparisons uninformative.

extracted by the encoder. We select 10 distinct observations from different states and replace their backgrounds with 40 unseen images. These images are then encoded using the learned representations from each algorithm. Observations corresponding to the same state are marked with identical colors. As illustrated in Figure 3, representations of images with varying backgrounds (distractors) but identical agent poses are embedded into tight clusters, indicating that the learned embeddings capture pose-invariant features. Notably, TRMS exhibits the most compact clustering and achieves the highest rewards under distracting backgrounds, suggesting a stronger ability to learn domain-invariant representations.

**Robustness to Distraction Levels.** Figure 4 illustrates the absolute difference in average rewards as the distraction level increases from video-easy to video-hard. The figure reveals that all methods experience considerable performance drops under higher distraction levels. Notably, TRMS exhibits the smallest fluctuation, closest to SGQN, while achieving considerable higher rewards than it, underscoring its resilience to elevated noise levels.

## 5.2 Locomotion

For Locomotion tasks, we utilise Unitree Series tasks (Hansen et al., 2021): Unitree Stand and Unitree Walk. The training and evaluation are performed in a similar way as described in Section 5.1 for DMC settings.

**Generalization Performance.** We evaluate generalization by running three seeds per task and computing the mean and standard deviation of returns. As shown in Table 2, TRMS outperforms the baselines across 3 out of 4 environments. In the video-easy setting, TRMS demonstrates a substantial 33.5% improvement in the Unitree Walk task, though it lags in Unitree Stand, resulting into lower average performance as compared to the baselines. However, as distractions intensify in the video-hard setting, TRMS consistently outperforms all baselines, showcasing remarkable resilience to distractor noise. Overall, TRMS achieves a 77.46% increase in performance in video-hard setting relative to the second-best method.

Table 2: Performance of various methods on Unitree Walk and Unitree Stand tasks for Video-Easy (VE) and Video-Hard (VH).

| Task (VE) | DrQ | DrQ-v2 | CURL | SVEA | SRM | PIEG | SGQN | TRMS (ours) |
|---|---|---|---|---|---|---|---|---|
| Walk | 67.4±9.2 | 97.8±15.7 | 74.8±14.2 | 98.4±28.3 | 98.0±9.4 | 140.2±63.9 | 151.7±87.1 | **202.6±47.2** (+33.55%) |
| Stand | 341.4±19.8 | 374.8±64.7 | 431.4±38.3 | **587.0±39.6** | 553.2±27.9 | 379.6±65.8 | 447.0±50.3 | 315.4±105.8 (-46.26%) |
| **Average** | 204.4 | 236.3 | 253.1 | 342.7 | 325.6 | 259.9 | 332.0 | **259.0** (-24.42%) |
| Task (VH) | DrQ | DrQ-v2 | CURL | SVEA | SRM | PIEG | SGQN | TRMS (ours) |
| Walk | 39.6±22.3 | 83.0±24.2 | 61.2±25.9 | 73.8±52.2 | 72.4±29.0 | 203.7±75.6 | 122.8±68.2 | **214.4±31.5** (+5.25%) |
| Stand | 65.6±25.7 | 95.8±37.4 | 99.4±25.3 | 279.3±10.7 | **300.0±34.5** | 202.0±43.1 | 139.8±47.0 | **305.4±101.7** (+1.80%) |
| **Average** | 52.6 | 89.4 | 80.3 | 176.6 | 186.2 | 202.8 | 131.3 | **259.9** (+77.46%) |

## 5.3 Dexterous Manipulation

Adroit (Rajeswaran et al., 2018) is an environment specifically designed for complex dexterous hand manipulation tasks, requiring substantial exploration and detailed feature extraction due to its sparse reward structure and the intricacy of its high-dimensional action space. In our experiments, we consider three of its tasks from a single view in RL-ViGen: Door, Hammer and Pen. This environment involves a magnitude of objects that needs to be masked, which makes the task extremely difficult.

TRMS achieved the highest average performance across tasks with scores of 59.5 in the video-easy setting and 54.9 in the video-hard setting. GQN matched TRMS at 59.5 in video-easy but scored lower in video-hard (30.3) due to heavy distractions. Notably, SGQN excelled on specific tasks like Door and Hammer, where TRMS's scores were comparatively lower, suggesting that while TRMS provides robust overall performance. However, there is a room for improvement in these environments. See the future directions below.

Table 3: Performance comparison of various methods on Adroit tasks with Video Easy (VE) and Video Hard (VH) background.

| Task (VE) | VRL3 | SVEA | SGQN | PIE-G | TRMS |
|---|---|---|---|---|---|
| Pen | 1.7±0.6 | 46.7±3.8 | 64.0±9.0 | 53.6±4.7 | **72.4±9.1** (+13.10%) |
| Door | 0.0±0.0 | 44.8±8.5 | **58.2±12.3** | 56.6±11.1 | 50.7±12.3 (-12.90%) |
| Hammer | 0.0±0.0 | 8.4±8.6 | **56.3±6.3** | 44.3±13.0 | 55.3±5.3 (-1.80%) |
| **Average** | 0.6 | 33.3 | **59.5** | 51.5 | **59.5** |
| Task (VH) | VRL3 | SVEA | SGQN | PIE-G | TRMS |
| Pen | 2.7±1.5 | 41.7±6.1 | 56.0±2.4 | 54.0±9.4 | **67.3±5.2** (+20.20%) |
| Door | 0.0±0.0 | 7.6±1.8 | 20.3±6.1 | **52.6±3.3** | 43.7±5.4 (-17.0%) |
| Hammer | 0.0±0.0 | 4.2±3.7 | 14.6±4.7 | 46.0±4.6 | **53.7±7.6** (+16.70%) |
| **Average** | 0.9 | 17.8 | 30.3 | 50.8 | **54.9** (+8.10%) |

## 6 Conclusion

To address the challenge of task-irrelevant distracting visual features in Visual Reinforcement Learning, we introduce TRMS, a method that utilizes existing masking strategies to extract masks from the visual scene. It segments and samples only the task-relevant masks. This approach eliminates the need for additional segmentation labels for individual tasks. We bypass the heavy computation time and resources by employing student network that learns these masks in few training steps. We evaluate TRMS on the RL-ViGen (Yuan et al., 2023) benchmark, covering tasks from the DeepMind Control Suite, Unitree locomotion, and Dexterous manipulation under varied distractions. TRMS achieves a 13.70% higher average reward in video-hard DeepMind tasks and surpasses baselines in 10 out of 12 tasks. It also yields a 77.46% improvement in locomotion tasks and comparable performance in dexterous manipulation, all while demonstrating robust resilience to increasing noise levels compared to baselines.

**Limitations and Future Directions.** Currently, the prior masking approach, FastSAM, generates masks independently for each state. Incorporating recent advancements like SAM2 (Ravi et al., 2024), which leverages temporal dependencies to refine mask extraction, could greatly enhance sampling efficiency. We need to select $k-$masks and initial teaching steps $T_{\text{Teach}}$ for distinct environments, depending on the complexity of the environment. An extension for automated selection of this hyperparameter would be extremely useful. Utilizing segmentation models with human-in-the-loop guidance via natural language descriptions (Zhang et al., 2024) can significantly enhance masking performance, especially in complex environments (Rajeswaran et al., 2018). Extending TRMS to non-agent-centric environments, such as CARLA, presents unique challenges, as masking-based methods often face limitations in these domains, warranting further investigation. Additionally, future research will explore the applicability of TRMS in more complex, real-world robotic applications, an area that remains largely underexplored in this domain.

## Acknowledgments

This work was created as part of the research projects 'KIRAMET' (FO999899661), MUTAVIA (FO999922732) and NNATT (FO999907606), which are funded by the Federal Ministry Republic of Austria Climate Action, Environment, Energy, Mobility, Innovation and Technology. It was also funded in part by the Austrian Science Fund (FWF) under grant 10.55776/COE12 (O. Özdenizci).

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

# A Implementation Details

We provide additional details on our implementation of TRMS. Table 4 summarizes the hyperparameters utilized in our study. For the DMC Generalization Benchmark (Hansen & Wang, 2021) and Locomotion Task (Hansen et al., 2021), we used the default hyperparameters specified in RL-ViGen (Yuan et al., 2023) for all the respective baselines.

Table 4: Hyperparameters for TRMS in DMC-GB.

| Hyperparameter | Values |
|---|---|
| Input size | $84 \times 84$ |
| Optimizer | Adam (Kingma, 2014) |
| Learning Rate (Actor,Critic, Masker) | $10^{-4}$ |
| Adam $\beta_1$, $\beta_2$ (All Networks) | 0.9, 0.999 |
| Discount ($\gamma$) | 0.99 |
| Frame Stack | 3 |
| Action Repeat | 2 |
| Initial Batch Size | 8 |
| Batch Size | 256 |
| Feature Dimension | 256 |
| Initial Sampling Steps | 2000 |
| Replay Buffer Size | 150K |
| Environment Steps | 500K |
| Gradient Steps Per Training Step | 1 |
| Target Update Interval | 2 |
| Target Smoothing Coefficient Critic | 0.01 |
| Target Smoothing Coefficient Encoder | 0.05 |
| Initial Temperature ($\alpha$ in SAC) | 0.1 |
| Temperature Learning Rate | $10^{-4}$ |
| Mask Sampler | AdamW (Loshchilov & Hutter, 2019) |
| Learning Rate | $10^{-5}$ |
| Number of Masks | 4 |
| Gumbel Softmax temperature ($\tau$) | 5.0 |
| Segmentation Model | FastSAM (Zhao et al., 2023) (default) |
| Image Size | 640 |
| IoU Threshold | 0.75 |
| Confidence Threshold | 0.40 |
| Overlap Mask | False |
| Initial Teaching Steps ($T_{\text{teach}}$) | 25K (DMC-GB) and 50K (Otherwise) |

The common Reinforcement Learning Hyperparameters were kept consistent with those used for TRMS, while method-specific parameters followed the configurations provided in their respective papers. Details on the specific hyperparameters are listed in Table 5.

Table 5: Hyperparameters for baselines in DMC-GB.

| Hyper-parameters | Value |
|---|---|
| Feature dim | DrQ-v2, CURL: 50; otherwise: 256 |
| N-step return | DrQ: 1; otherwise: 3 |
| Optimizer | Adam |
| Hidden dim | 1024 |
| Frame stack | 3 |
| SGQN Quantile Threshold | 0.95 or 0.98 |
| Critic Weight Decay | $10^{-5}$ |
| SGQN Auxiliary Learning Rate | 8e-5 |

For the Dexterous Manipulation tasks, we use the same hyperparameters utilised for the respective baselines as mentioned in their papers and for environments as mentioned in RL-ViGen (Yuan et al., 2023). They are described in the Table 6.

Table 6: Adroit Hyperparameters.

| Hyper-parameter | Task | Value |
|---|---|---|
| Training Frames | Hammer | $10^6$ |
| | Door | $10^6$ |
| | Pen | $2 \times 10^6$ |
| Learning Rate | Hammer | $10^{-4}$ |
| | Door | $10^{-4}$ |
| | Pen | $10^{-4}$ |
| $k-$Masks | Hammer | 15 |
| | Door | 15 |
| | Pen | 10 |
| SGQN Quantile | Hammer | 0.9 |
| | Door | 0.9 |
| | Pen | 0.9 |
| SGQN Critic Weight | Hammer | 0.9 |
| | Door | 0.5 |
| | Pen | 0.9 |
| SGQN Auxiliary Learning Rate | Hammer | $8 \times 10^{-5}$ |
| | Door | $8 \times 10^{-5}$ |
| | Pen | $8 \times 10^{-5}$ |

## B  Soft Actor-Critic Algorithm

The Soft Actor-Critic (SAC) (Haarnoja et al., 2018) algorithm is an off-policy reinforcement learning method that maximizes cumulative rewards while promoting entropy to encourage exploration. The objective for SAC is given by

$$\pi^* = \arg\max_{\pi_{\phi_a}} \mathbb{E}_{(s_t, a_t) \sim \pi_{\phi_a}} \left[ \sum_{t=0}^{T} r(s_t, a_t) + \alpha \mathcal{H}(\pi_{\phi_a}(\cdot | s_t)) \right], \tag{11}$$

where $\alpha$ is a temperature parameter that balances reward and entropy terms. SAC employs two critic networks, $Q_{\theta_{q_1}}$ and $Q_{\theta_{q_2}}$, trained to minimize the soft Bellman residual

$$L(\theta_{q_i}) = \mathbb{E}_{(s,a,r,s')\sim\mathcal{D}} \left[ \left( Q_{\theta_{q_i}}(s,a) - \left( r + \gamma \, \mathbb{E}_{a'\sim\pi_{\phi_a}} \left[ \min_{j=1,2} Q_{\theta_{q_j}}(s',a') - \alpha \log \pi(a'|s') \right] \right) \right)^2 \right]. \tag{12}$$

The actor network $\pi_{\phi_a}$ is updated to maximize the expected Q-value regularized by an entropy term, defined as

$$\mathcal{L}_\pi(\phi_a) = \mathbb{E}_{s\sim D, a\sim\pi_{\phi_a}} \left[ \alpha \log \pi_{\phi_a}(a|s) - Q_{\theta_q}(s,a) \right], \tag{13}$$

where $\alpha$ is the entropy temperature coefficient. The corresponding policy gradient is given by

$$\nabla\mathcal{L}_\pi(\phi_a) = \mathbb{E}_{s,a} \left[ \nabla_{\phi_a} \log \pi_{\phi_a}(a|s) \left( \alpha - Q_{\theta_q}(s,a) \right) \right]. \tag{14}$$

To balance exploration and exploitation, SAC adapts $\alpha$ by minimizing the temperature objective:

$$\mathcal{L}(\alpha) = \mathbb{E}_{a\sim\pi_{\phi_a}} \left[ -\alpha \log \pi_{\phi_a}(a|s) - \alpha\mathcal{H} \right], \tag{15}$$

where $\mathcal{H}$ denotes the target entropy, encouraging diverse action sampling in high-dimensional action spaces.

## C   Architecture Details

**Encoder.** The encoder network consists of two main components: a shared convolutional module and a subsequent linear projection. The shared convolutional module is an 11-layer network designed to process input observations composed of 3 stacked RGB frames, with dimensions $[9, 84, 84]$, ultimately generating spatial feature maps. The first layer employs a $3 \times 3$ convolutional kernel with a stride of 2 and 32 output channels, allowing for an early reduction in spatial resolution. The remaining layers are structured as sequential ReLU-convolution blocks, each composed of a ReLU activation followed by a $3 \times 3$ convolution with a stride of 1 and maintaining 32 channels across all layers. This uniform channel depth preserves consistency in feature representation throughout the network.

The final convolutional output is then flattened into a feature vector of size $32 \times 21 \times 21$. This vector is subsequently passed through a linear projection layer, which reduces the dimensionality to 512, thus producing a condensed latent representation suitable for further processing. This final representation serves as the input to the policy and value networks, allowing for efficient and effective state encoding. Furthermore, input frames are normalized by scaling to the range $[-0.5, 0.5]$

**Actor.** The actor network comprises a feature extractor and a policy head. The feature extractor maps the 512-dimensional input representation to a 256-dimensional latent space via a fully connected layer, followed by layer normalization and Tanh activations for normalized, non-linear transformations. The resulting features are then passed through two hidden layers with 1024 units and ReLU activations, capturing complex action-value mappings.

The final layer outputs action means, $\mu$, which are scaled by a Tanh activation to enforce bounded action outputs. The standard deviation, $\sigma$, is constant and scaled by an input parameter, std. Together, $\mu$ and $\sigma$ parameterize a Truncated Normal distribution for continuous action sampling.

**Critic.** The critic network comprises two parallel Q-networks, $Q_1$ and $Q_2$, which are employed to estimate state-action values and mitigate overestimation bias. Initially, observations are passed through a shared feature extraction module that projects the input representation of dimension 512 into a 256-dimensional feature vector. This transformation is accomplished through a fully connected layer, followed by layer normalization and Tanh activation, which ensures normalized outputs and reduces the likelihood of activation saturation.

The resulting 256-dimensional feature vector is concatenated with the action input, forming a joint representation that is processed by each Q-network independently. Both $Q_1$ and $Q_2$ are structured with two hidden layers, each containing 1024 units, and use ReLU activations to introduce non-linearity, enabling the networks to model complex value functions effectively. The final layer of each Q-network outputs a single scalar, representing the $Q$-value for the given state-action pair. By using two independent Q-networks, the critic can take the minimum of both $Q$-value estimates, which reduces overestimation—an issue commonly encountered in value-based reinforcement learning. This design contributes to the stability and robustness of the learned policy.

**Mask Sampler.** The Mask Sampler network is designed to process multiple input masks and output selection probabilities for each. A single input has the shape of $(k, 84, 84)$, where $k$ is the number of masks. The network comprises three convolutional layers, each with $3 \times 3$ kernels and a padding of 1 to maintain spatial dimensions. The first layer maps the input channels, corresponding to the number of masks, to 32 feature maps, followed by batch normalization and ReLU activation. The feature depth is then sequentially increased to 64 and 128 channels by the subsequent convolutional layers, each followed by batch normalization and ReLU to enhance spatial feature extraction.

The output of the final convolutional layer undergoes global average pooling to reduce the spatial dimensions to $1 \times 1$, yielding a 128-dimensional vector. This vector is then passed through a fully connected layer with 64 units and a ReLU activation, followed by a final linear layer that outputs logits corresponding to the number of masks. A Gumbel-Softmax activation with subsequent temperature is applied to these logits.

**Masker.** The Masker network is a convolutional architecture designed to produce a single-channel mask from RGB inputs. The network consists of five convolutional layers, each with a $3 \times 3$ kernel and a padding of 1 to maintain spatial dimensions. The first two layers map the input image, with three color channels, to 64 feature channels through successive applications of convolution, ReLU activation, and Batch Normalization, promoting feature extraction while stabilizing training.

The third and fourth layers reduce the feature depth to 32 channels, employing similar ReLU and batch normalization operations to preserve spatial information while refining feature representations. The final convolutional layer outputs a single-channel feature map, which represents the generated mask.

The output feature map is then passed through a sigmoid activation function, designed to constrain output values between 0 and 1. All convolutional layers are initialized with Xavier uniform initialization, ensuring balanced weight distributions. This architecture allows Masker to effectively learn spatial patterns for producing accurate binary masks from RGB input data.

## D Wall Time

As TRMS incorporates both SAM and a student network to enhance computational efficiency, here we compare the wall time required for convergence between TRMS and the SAM-only variant. Evaluation is performed across three seeds of the *Walker Walk* task, where the teacher network is employed for the first 25k steps.

As shown, TRMS achieves an average wall time of approximately **15 hours**, whereas the SAM-only variant reaches up to **3 days and 1 hour** ($\sim$73 hours). This substantial reduction demonstrates that TRMS not only accelerates learning early on but also sustains faster convergence through its student network (due to its small architecture), which absorbs knowledge from the heavy parameterised teacher network.

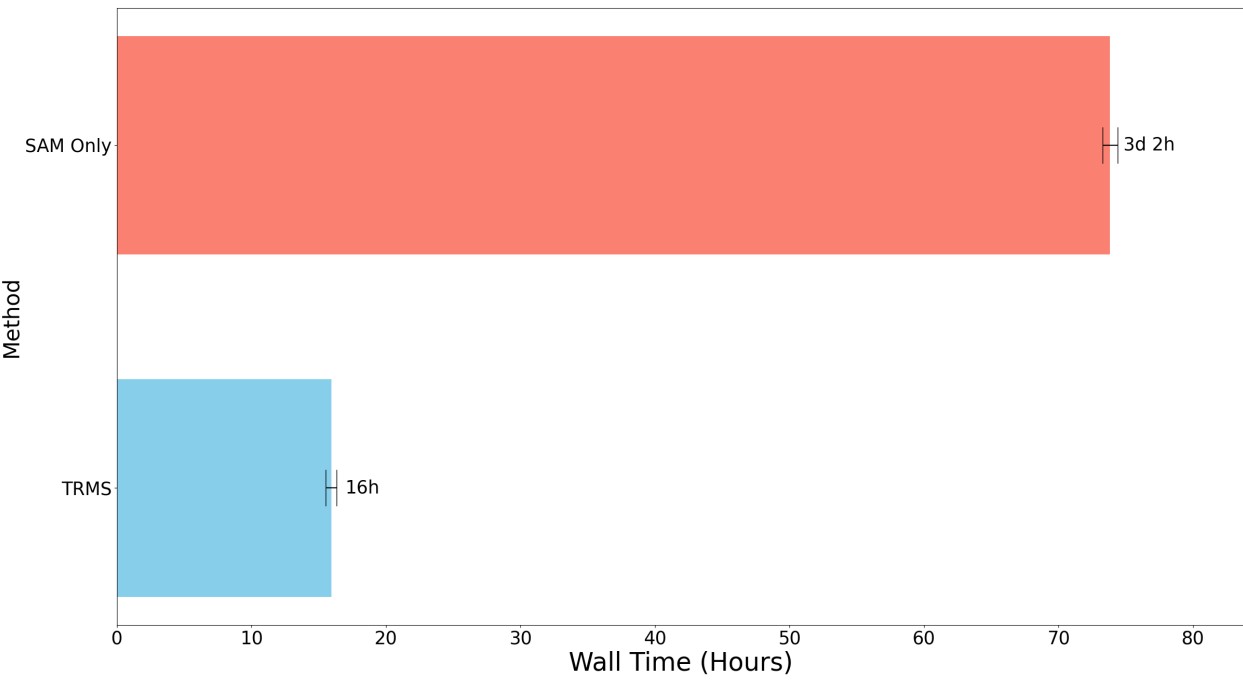

Figure 5: Comparison of TRMS and SAM-Only Wall time for Walker Walk on three seeds.

## E    Additional Visual Results

The following visual demonstrations showcase selected frames from the video-hard environment. On the left, each image displays the actual observation, while the corresponding masked observation is presented on the right. These comparisons highlight the complexity of the environment, where the masking process isolates relevant features, facilitating the agent's focus on relevant task elements amidst challenging distractors.

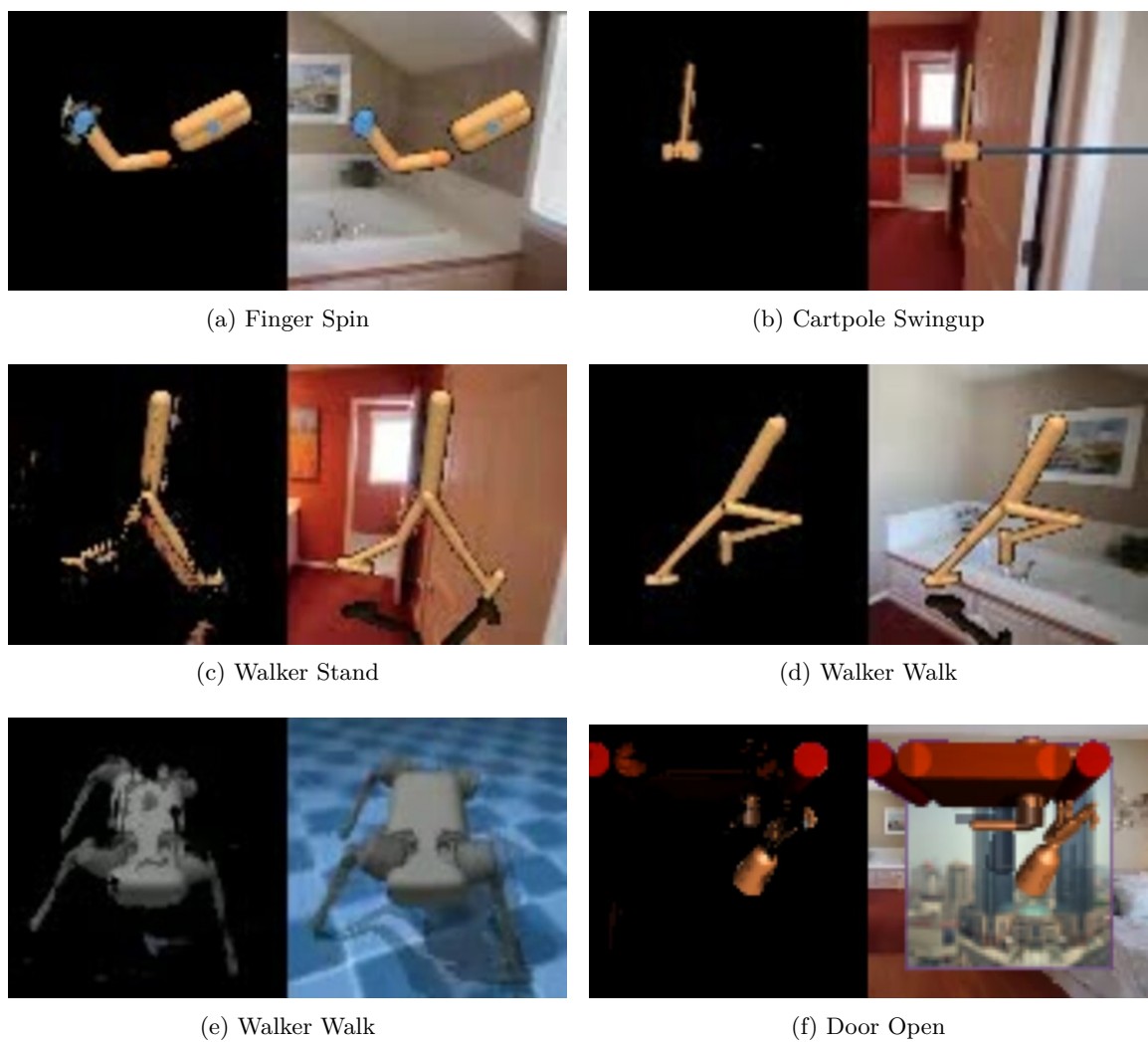

(a) Finger Spin                         (b) Cartpole Swingup

(c) Walker Stand                         (d) Walker Walk

(e) Walker Walk                         (f) Door Open

Figure 6: Visual examples from the video-hard environment, showing actual observations (left) and corresponding masked observations (right), highlighting relevant feature isolation.

## F    Baseline Results:

Baseline results presented in this paper were obtained as follows. For the DMC-GB environments, we re-implemented most baselines ourselves, with the exceptions of DrQ, DrQ-v2, and SVEA, whose performance numbers were directly cited from the PIE-G paper (Yuan et al., 2022b). For the PIE-G baseline, we conducted experiments using the authors' original implementation. SGQN (Bertoin et al., 2022) results were reproduced using the official implementation provided with the RL-ViGen (Yuan et al., 2023) benchmark. For Locomotion and Manipulation environments, all baseline performances are directly cited from the RL-ViGen benchmark [3].

---

[3]https://github.com/gemcollector/RL-ViGen/tree/master/results

