# OpenReview forum: "Learning Robust Representations for Visual Reinforcement Learning via Task-Relevant Mask Sampling"
_TMLR — Accepted by TMLR_

### Review · Reviewer_n92v · 2025-06-15

**Summary Of Contributions:**

This paper proposes a novel method, Task-Relevant Mask Sampling (TRMS), to improve the generalization and robustness of visual reinforcement learning (RL) agents in environments with high-dimensional observations and severe background distractions. The main contributions of this work are as follows:

1. Task-Relevant Masking Framework:
The authors introduce TRMS, which leverages pre-trained image segmentation models (such as FastSAM) to generate multiple candidate object masks for each observation. A mask selection network then dynamically selects the subset of masks most relevant to the task at each timestep, allowing the RL agent to focus on task-relevant visual regions and filter out distractors.


2. Efficient Student Network Distillation:
To address the computational overhead of running segmentation models throughout training, the authors develop a lightweight student network. This network is trained to imitate the masking process of the pre-trained segmentation model during an initial phase and subsequently replaces the heavy segmentation module for the remainder of training, greatly improving computational efficiency.


3. Comprehensive Empirical Evaluation:
The method is evaluated on a suite of challenging vision-based RL benchmarks (RL-ViGen), including variants of DeepMind Control Suite, Quadruped Locomotion, and Dexterous Manipulation with video-based background distractions. TRMS demonstrates superior generalization and robustness compared to a broad set of prior methods, achieving state-of-the-art performance in most settings, especially under high-noise conditions.


4. Ablation and Analysis:
The paper includes visualizations and ablation studies to illustrate the effectiveness of mask selection and the learned representations' invariance to visual distractors.



Overall, the work presents a practical and effective framework for isolating task-relevant information in visual RL, and sets a new benchmark for robust generalization in distraction-rich environments.

**Audience:**

Yes

**Claims And Evidence:**

Yes

**Requested Changes:**

Recommendation for Improvement:
The paper would benefit significantly from a detailed clarification of how the baseline results were obtained. If the baselines were re-implemented, please specify the exact experimental settings and confirm that they match those of the original papers. If the numbers are cited, please clearly indicate the source and discuss any known discrepancies. Providing open-source code and/or additional ablation studies to support reproducibility would also greatly strengthen the contribution.

**Strengths And Weaknesses:**

Strengths

Clear Motivation and Human-Inspired Perspective:
The paper addresses an important challenge in visual reinforcement learning—robustness and generalization in noisy, high-dimensional environments—by leveraging human-like segmentation and attention mechanisms. This conceptual approach is well-motivated and grounded in both cognitive science and machine learning literature.

Comprehensive Related Work and Experimental Scope:
The literature review is thorough, and the comparisons cover a wide range of recent and relevant baseline methods. The experimental evaluation spans multiple well-known benchmarks and difficult tasks, making the results broadly credible and informative.



---

Weaknesses

Unclear Baseline Numbers and Potential Reproducibility Issue:
The most significant concern is the lack of clarity regarding the source of the baseline results reported in the experimental tables. It is not explicitly stated whether these baseline numbers are reproduced by the authors under consistent experimental settings, or if they are taken directly from the original papers.
In several cases, the baseline performances (e.g., for DrQ, SVEA, SRM, PIEG, SGQN) in challenging settings appear substantially lower than those reported in the respective original publications. This discrepancy raises concerns about the experimental fairness and reproducibility of the results.
Without a clear explanation, it is difficult to judge whether the observed improvements stem from genuine methodological advances, or from potential inconsistencies in baseline reproduction (e.g., differences in hyperparameters, training steps, seeds, or evaluation protocols).

---

> ### Author Response · Authors · 2025-07-06
> **Response to Reviewer n92v**
>
> We sincerely thank the reviewer for their thorough reading of our manuscript and for providing insightful and constructive feedback. Your detailed comments and suggestions have been invaluable in improving the quality and clarity of our work. We have addressed the concerns below.
>
> > *The paper would benefit significantly from a detailed clarification of how the baseline results were obtained. If the baselines were re-implemented, please specify the exact experimental settings and confirm that they match those of the original papers. If the numbers are cited, please clearly indicate the source and discuss any known discrepancies.*
>
> We thank the reviewer for raising this important clarification. For the DMC-GB environments, we re-implemented several baseline methods ourselves, except for DrQ, DrQ-v2, and SVEA, whose results were directly cited from the PIE-G paper. For PIE-G itself, we conducted our own experiments using the original implementation provided by the authors and achieved comparable or better results in some scenarios (Video Hard: Walker Walk- Ours: 641, Pie-G Paper: 600). For SGQN, we executed the official code provided by the RL-ViGen authors, utilizing the same hyperparameters from the original paper, ensuring consistency with the established evaluation protocol.\
> For other benchmark environments, specifically Locomotion and Manipulation tasks, baseline results were taken directly from the RL-ViGen benchmark. We clearly specify these sources in the **Appendix (Section F)** to ensure transparency and facilitate reproducibility. We will also release the checkpoints with our code for transparency.

---

> > ### Comment · Reviewer_n92v · 2025-07-06
> >
> > Thanks for your response. I will look over the revised parts soon to verify the claims and evidence.

---

### Review · Reviewer_MqPz · 2025-06-24

**Summary Of Contributions:**

The TRMS method introduces a masking technique that identifies and preserves task-relevant image regions, leading to enhanced generalization and robustness in noise-prone environments. Furthermore, the authors train a student network to reduce computational complexity of FastSAM. Finally,the authors showcase the performance of their method in a number of diverse simulation environments and they compare it with a number of baselines.

**Audience:**

Yes

**Claims And Evidence:**

Yes

**Requested Changes:**

Kindly address the weaknesses listed above.

**Strengths And Weaknesses:**

Strengths:
The paper is well written and it proposes TRMS, a computationally efficient method for segmentation-driven masking in RL, leveraging FastSAM and a student network. Furthermore, the  authors provide a  substantial comparison with baselines to showcase the performance of TRMS.

Weaknesses:
How is hyperparameter tuning done in the paper? The method requires selecting the number of masks and the duration of the teaching phase, which are environment-dependent. Is there systematic way to choose those hyperparameters

It is not clear where the loss $\mathcal{L}_{\pi}$ in algorithm 1 is defined

Why is there no comparison with FTD and MaDi? They seem to be closely related methods. If there is a particular reason, then it should be mentioned otherwise they should be part of the baselines.

The experiments use low-resolution inputs. It is unclear how TRMS would scale to high-resolution observations, where segmentation and masking could become computationally prohibitive.

How is diversification ensured when training the student network and how is over-fitting avoided? This part of the paper needs some more clarity.

$\mathcal{L}_{masker}$ appears before it is defined.

---

> ### Author Response · Authors · 2025-07-06
> **Response to Reviewer MqPz**
>
> We sincerely thank the reviewer for their thorough reading of our manuscript and for providing insightful and constructive feedback. Your detailed comments and suggestions have been invaluable in improving the quality and clarity of our work. We have addressed the concerns below.
>
> > *How is hyperparameter tuning done in the paper? The method requires selecting the number of masks and the duration of the teaching phase, which are environment-dependent. Is there systematic way to choose those hyperparameters?*
>
> We appreciate the reviewer’s emphasis on hyperparameter selection. In our experiments, the number of masks and the teaching phase duration were selected heuristically based on initial observations about task complexity. Specifically, environments with more visual complexity and distractors naturally benefit from a longer teaching phase and more masks. While systematic hyperparameter selection methods such as Grid Search or Bayesian optimization could further enhance performance, our empirical results demonstrate that the chosen hyperparameters yield consistent and robust performance across multiple tasks.
>
> > *It is not clear where the loss $\mathcal{L}_{\pi}$ in algorithm 1 is defined.*
>
> We have changed the Section B of the Appendix to make it more clear and explicit.
>
> > *Why is there no comparison with FTD and MaDi? They seem to be closely related methods. If there is a particular reason, then it should be mentioned otherwise they should be part of the baselines.*
>
> We thank the reviewer for pointing out the importance of comparisons with FTD and MaDi. Regarding FTD, we explicitly discussed its methodological and computational limitations in Section 2 (Related Work), highlighting that its continuous reliance on SAM throughout training significantly increases computational overhead, making it impractical for the multi-object scenarios we consider. MaDi, while relevant, was not included as it is not part of the benchmark (RL-ViGen) we utilized. We have considered many other baselines which performs exceptionally well in these cases. Instead, we focused on a broad set of strong and widely recognized baselines within RL-ViGen, many of which demonstrate state-of-the-art performance under challenging visual distraction settings.
>
> > *The experiments use low-resolution inputs. It is unclear how TRMS would scale to high-resolution observations, where segmentation and masking could become computationally prohibitive.*
>
> The reviewer raises an important scalability concern. Although our primary experiments used standard-resolution inputs commonly employed in visual RL benchmarks (e.g., 84×84 pixels), the TRMS framework is explicitly designed to mitigate scalability issues. By replacing the computationally expensive SAM model with a lightweight student network early during training,x TRMS significantly reduces the computational cost associated with segmentation-based masking. While higher resolutions would indeed increase computational demands, the two-stage masking strategy ensures practical scalability compared to methods continually relying on SAM.
>
> > *How is diversification ensured when training the student network and how is over-fitting avoided? This part of the paper needs some more clarity.*
>
> We thank the reviewer for highlighting this important point. Diversification during student network training is ensured primarily through standard data augmentation techniques which significantly increase variability in the training data and reduce susceptibility to overfitting. Similar strategies have been widely validated in previous works like SVEA, PIE-G etc. Additionally, the student network is explicitly optimized using task-specific Q-value losses rather than reconstruction-based objectives, inherently encouraging it to capture generalizable mask-selection patterns rather than memorizing task-specific or instance-specific state details.
>
> > *$\mathcal{L}_{\text{masker}}$ appears before it is defined.*
>
> We have changed the main text to refer to equation 10.

---

### Review · Reviewer_qsNd · 2025-06-24

**Summary Of Contributions:**

Please check the strengths.

**Audience:**

Yes

**Broader Impact Concerns:**

N\A

**Claims And Evidence:**

Yes

**Requested Changes:**

Please check the weaknesses.

**Strengths And Weaknesses:**

## Strengths:
- This paper targets at an essential problem - improving the robustness and generalization of visual reinforcement learning.
- The empirical results are sufficient by involving multiple benchmarks and baselines. The proposed algorithm demonstrates state-of-the-art performance.

## Weaknesses:
- The novelty is limited, which combines SAC and SAM through a mask selector.
- There should be a comparsion with related works for generalization in visual RL in Section 2, to showcase the novelty and research gap.
- It would be important to explicitly testifiy the following statement through illustrative exmaples: "This occurs because selecting an irrelevant segment means that, during each state sampling from the replay buffer, an insignificant pixel in a given state might assume varying values, despite the pixel contributing nothing meaningful to the task value. As a result, the mask selection process is encouraged to discard these irrelevant pixels. ", as this is the core motivation.
- The baselines are relatively old - from 2022 and before.

---

> ### Author Response · Authors · 2025-07-05
> **Response to Reviewer qsNd**
>
> We would like to express our sincere gratitude for the suggestions and careful readings of the reviewer. Thank you for appreciating our contribution. We have revised the manuscript and will address your questions below. The revision in the manuscript is highlighted in blue.
> > *The novelty is limited, which combines SAC and SAM through a mask selector.*
>
> Thank you for raising this point. While our method does indeed integrate SAC with SAM through a mask selector, we emphasize that our primary novelty lies precisely in this integration, leveraging SAM explicitly for mask-based selective attention in reinforcement learning contexts. Previous works typically employed fixed masks or heuristic attention mechanisms, lacking the simple selection mechanism from the masks provided by the SAM. By introducing a learned mask selector, our method dynamically identifies task-relevant segments, substantially improving robustness under visual distractions, which, to our knowledge, has not been explored before.
>
> > *There should be a comparison with related works for generalization in visual RL in Section 2, to showcase the novelty and research gap.*
>
> We appreciate this suggestion. We have expanded Section 2 (Related Works) to explicitly position our contribution against recent methods addressing generalization in visual reinforcement learning. The added comparisons, highlighted in blue for clarity, underscores the research gap and novel contributions relative to existing methods.
>
> > *It would be important to explicitly testify the following statement through illustrative examples: "This occurs because selecting an irrelevant segment means that, during each state sampling from the replay buffer, an insignificant pixel in a given state might assume varying values, despite the pixel contributing nothing meaningful to the task value. As a result, the mask selection process is encouraged to discard these irrelevant pixels," as this is the core motivation.*
>
> We thank the reviewer for highlighting this point. While we acknowledge that additional visual illustrations could further strengthen the explanation, we believe the mechanism is already supported by the observed performance improvements in high-distractor settings (Table 2, Section 5 and Figure 6 in Section E). These results demonstrate that the mask selector effectively suppresses irrelevant regions, leading to more stable and task-relevant representations. Furthermore, this behavior is conceptually consistent with prior works (e.g., SGQN [1], Achille and Soatto[2]) that argue for filtering out uninformative features to improve policy robustness. To address this concern, we propose revising the original statement for clarity and conciseness as follows: \
> *By discarding visually irrelevant segments, the mask selector reduces representational variance from distractor pixels and encourages the policy to focus on task-relevant features.*\
> We hope this rephrasing and our existing results together sufficiently clarify the intended intuition.
>
> > *The baselines are relatively old - from 2022 and before.*
>
> We appreciate the reviewer highlighting this point. Indeed, the selected baselines primarily include methods proposed up to 2022; however, these were intentionally chosen as they constitute the complete set of methods available within the adopted benchmark (RL-ViGen) and are among the best-performing methods established in the literature. This selection ensures rigorous, fair, and meaningful comparisons against robust and well-recognized approaches.
>
>
> **References:**
>
> [1] Bertoin, David, et al. "Look where you look! Saliency-guided Q-networks for generalization in visual Reinforcement Learning." Advances in neural information processing systems 35 (2022): 30693-30706.
>
> [2] Achille, Alessandro, and Stefano Soatto. "Emergence of invariance and disentanglement in deep representations." Journal of Machine Learning Research 19.50 (2018): 1-34.

---

### Decision · Action_Editor_vefS · 2025-08-11

**Recommendation:** Accept with minor revision

**Additional Comments:**

The authors may add an explanation for $m^\times$ in the third row of figure 1.

**Audience:**

Yes

**Audience Explanation:**

This work will be of interest to researchers working on reinforcement learning, particularly on robustness under noisy settings.

**Claims And Evidence:**

Yes

**Claims Explanation:**

This paper studies visual reinforcement learning in noise-saturated environments. The authors propose a new method, Task-Relevant Mask Sampling (TRMS), which integrates the Segment Anything Model with Soft Actor-Critic to improve generalization and robustness. To address the computational overhead of running segmentation models throughout training, the authors develop a lightweight student network.

The proposed method TRMS is evaluated on a suite of challenging vision-based RL benchmarks, including variants of DeepMind Control Suite, Quadruped Locomotion, and Dexterous Manipulation with video-based background distractions.  The results demonstrates superior generalization and robustness of TRMS compared to a broad set of prior methods, achieving state-of-the-art performance in most settings—particularly under high-noise conditions (Table 1). Overall, all reviewers agree that the performance claims for the proposed method are supported by the experimental results.

Reviewers raised several concerns about the experimental setup, including hyperparameter tuning and baseline selection, which are addressed in the rebuttal.